# Fibromyalgia Detection Based on EEG Connectivity Patterns

**DOI:** 10.3390/jcm10153277

**Published:** 2021-07-25

**Authors:** Ramón Martín-Brufau, Manuel Nombela Gómez, Leyre Sanchez-Sanchez-Rojas, Cristina Nombela

**Affiliations:** 1Unidad de Corta Estancia, Hospital Psiquiátrico Román Alberca, National Service of Health, 30120 Murcia, Spain; r.martinbrufau@um.es; 2Research Support Service, University of Murcia, 30100 Murcia, Spain; manolonombela@gmail.com; 3Regenerative Medicine and Advanced Therapies Lab., Instituto de Investigación Sanitaria San Carlos (IdIISC), Hospital Clínico San Carlos, 28040 Madrid, Spain; leyre.sanchez@salud.madrid.org; 4Biological and Health Psychology, Autonomous University of Madrid (UAM), 28049 Madrid, Spain

**Keywords:** fibromyalgia, EEG, fast Fourier transform, diagnosis, ROC curve

## Abstract

Objective: The identification of a complementary test to confirm the diagnosis of FM. The diagnosis of fibromyalgia (FM) is based on clinical features, but there is still no consensus, so patients and clinicians might benefit from such a test. Recent findings showed that pain lies in neuronal bases (pain matrices) and, in the long term, chronic pain modifies the activity and dynamics of brain structures. Our hypothesis is that patients with FM present lower levels of brain activity and therefore less connectivity than controls. Methods: We registered the resting state EEG of 23 patients with FM and compared them with 23 control subjects’ resting state recordings from the PhysioBank database. We measured frequency, amplitude, and functional connectivity, and conducted source localization (sLORETA). ROC analysis was performed on the resulting data. Results: We found significant differences in brain bioelectrical activity at rest in all analyzed bands between patients and controls, except for Delta. Subsequent source analysis provided connectivity values that depicted a distinct profile, with high discriminative capacity (between 91.3–100%) between the two groups. Conclusions: Patients with FM show a distinct neurophysiological pattern that fits with the clinical features of the disease.

## 1. Introduction

Fibromyalgia (FM) is a highly prevalent, painful disease, suffered by 2–4% of the population in the industrialized world, predominantly in women (ratio 9:1); it is very debilitating both physically and psychologically [1]. Current diagnosis criteria evaluate neither peripheral nor central functional deficiencies linked with the clinical symptoms, which complicates both the identification of the physiopathology of the disease and the search for adequate treatment [2]. 

In general terms, FM represents an enormous expenditure of resources, both direct (health care and medication) and indirect (e.g., loss of jobs and use of government aid) for the health, social and economic systems. The mean annual cost per patient in western countries ranged from US $2274 to $9573 in the central studies and even more in others, depending on the severity of symptoms and methods of cost calculation [3]. There exists, therefore, a need to identify a discriminative complementary method which, together with the description of the clinical symptoms, would help in the detection of FM [2]. Early diagnosis and treatment would reduce the burdens on patients, relatives, and society.

The feeling of pain is generated by a widely distributed brain network rather than by a direct sensory input evoked by a lesion or other pathology [4]. There are clear, substantial differences between (i) acute pain, which is evoked by specific noxious inputs, and whose sensory transmission mechanisms are well described, and (ii) chronic pain syndromes, which are often characterized by severe pain associated with little or no discernible injury or pathology, and are still not well understood [4]. In the last five years, some studies have shed some light on the cerebral mechanisms of chronic pain modulation [5], showing that subjective pain experience corresponds to a defined pattern of brain activity, in what is called the ‘pain matrix’ [6,7]. 

Such experience of pain has consequences that go further than temporary unpleasant feelings. Pain leaves a footprint: experiencing chronic pain can cause anatomical and functional reorganization of the brain [8], as shown in several disorders such as phantom pain, chronic back pain, irritable bowel syndrome, and FM (May, 2008). The morphological changes include the loss of gray matter volume [9], whereas the functional alterations include aberrant functional activity [10]. Recent reviews of the long-term effects of chronic pain have demonstrated the appearance of a “brain signature” [7,11]. The neural dynamics of the experience of pain follow specific processes related to coherence, activation, and deactivation of “core structures” in the default mode network, which are not linearly associated with stimulus intensity [12]. Such oscillations and the synchrony characteristics of pain can be measured by EEG and studied [13,14], including in clinical settings [15]. For example, Jensen et al. (2013) reported that the presence of specific EEG patterns might predict with 83% accuracy spinal cord injury patients’ vulnerability to feelings of chronic pain [16], and Vuckovic et al. (2018) reported up to 85% accuracy in other studies [17]. A similar phenomenon takes place in healthy controls [15]. 

In the case of FM generalized pain, allodynia, and other neurological symptoms of central origin [18], result from deep tissue, like joints or muscles, in combination with central sensitization mechanisms. The nociceptive input may begin in peripheral tissue (e.g., by infection), generating allodynia or central sensitization. Those impairments in pain mechanisms might derive from long-term neuroplastic imbalances that the patients’ antinociceptive capabilities cannot manage and result in ever-increasing pain sensitivity and dysfunction [19].

The physiopathology of the FM syndrome described in the literature is compatible with a central state of hyperexcitability of the nociceptive system [20,21], specifically, a persistent over-activation of Theta and Beta bands [22,23].

Complementary research indicates that both power band and power density differences exist between patients and matched controls. For example, Delta power density at temporal areas appears to be decreased in patients concerning controls, but Beta power reaches higher values in frontal and cingulate regions of patients with FM (Gonzalez-Roldan). Other types of analysis based on EEG data as source analysis have also indicated differences in the cingulate cortex. In particular, Vanneste et al. focused on both the degree of activation and the degree of integration and concluded that patients with FM showed decreased connectivity in the anterior cingulate cortex, which may affect the pain inhibitory pathway, mediating the pain feeling [24]. Pain experience correlates with a decreased level of Alpha-2 (11–12 Hz) in the posterior cortex [25] and decreased connectivity of the insula within the default mode network [26].

Other neuroimaging techniques, such as magnetoencephalography (MEG), have also shown disrupted connectivity at the Theta frequency for patients with FM compared with controls in the default mode network [27] and with resting state sequences [28]. However, there is little information comparing patients with FM with controls on electrical coherence between brain regions [24], and specific connections between regions have not yet been addressed. The analysis of such differences could, perhaps, be based on the EEG technique [29]. Working on EEG data, this study aims to describe the connectivity patterns in the default mode network in a group of patients to establish differential parameters and compare them with a sample of healthy controls. The hypothesis is that patients with fibromyalgia present significant differences from the controls in EEG activity and that this differential activity would be linked to a decrease in brain connectivity, as happens in other diseases related to chronic pain [30,31,32,33].

## 2. Materials and Methods

### 2.1. Sample

The sample consisted of 23 patients with FM and 23 healthy control subjects. The patients with FM were recruited via FIBROFAMUR (the association of patients with FM). They had all been diagnosed with FM by the Rheumatology Department of the Virgen de la Arrixaca University Hospital, Murcia) according to the American College of Rheumatology Diagnostic Criteria for fibromyalgia, with no record of epilepsy seizures or other neurological disorder in the sample. Recruiting, testing, and analysis of the data took place in 2017 and 2019. 

The two groups were matched for age and gender. The mean age of the experimental group was 56 years (range 35–65). Seventy-seven percent had had the disease for more than 12 years. T-test analysis indicated no significant differences in age (t (44) = 0.061, *p* = 0.54) with the control group. The gender distribution was five males (21.8%) and eighteen females (78.2%) in each group. The control sample was randomly collected from the PhysioBank database, a public standard database available on the internet from the National Center for Research Resources of the National Institutes of Health [34]. It came from an experiment on motor movement/imagery, with a baseline of eyes closed EEG in resting state conditions, similar to the FM patients. A history of neurological or neuropsychiatric records or the use of drugs were exclusion criteria for participation in the control group. Accordingly, no pathological signals were found on the EEG registry of the control sample. Control sample registries undergo visual inspection to check the absence of pathological signs [35]. 

The Ethical Committee of the University of Murcia (Spain) approved the study. All procedures used followed the ethical standards of the responsible committee on human experimentation (institutional and national) and with the Helsinki Declaration of 1964 and its later amendments. All patients signed a written informed consent before their inclusion in the study.

### 2.2. Procedure

Twenty-one high-resolution EEG channels (NEURON-SPECTRUM-AM^®^) were used to record the data following 15 min eyes-closed resting state protocol. The sampling rate was 512 Hz. Guidelines to standard sample registration were followed (demographics, medical history, screening EEG, medication status) [36]. Data acquisition was conducted in complete silence, sitting in a comfortable and isolated room free from unpleasant stimuli. The disposition of electrodes followed the international 10–20 system with earlobes used as references and ground reference located in the Fpz location.

#### 2.2.1. Data Preprocessing

Average reference, filtering, and analysis of the EEG signal were equal in both samples. A medical doctor with 50 years’ experience in EEG ran a qualitative analysis by visual inspection of specific assemblies of bipolar montages. Segments of no less than 1 min of EEG free from artifacts were selected for analysis. The grapho-elements of pathological significance were registered (i.e., frequent posterior sharp waves) [37] since the morphological mapping tests of these patients usually show the presence of areas with atrophy [38,39], with signs of aging [40] or irrigation alterations [41] (Figure 1). 

#### 2.2.2. Data Analysis

Global EEG power was calculated with a weighted average across all channels with an average reference. Quantitative analyses (both fast Fourier transformation (FFT) and coherence analysis) were performed using the Brainstorm software [42] on MATLAB compiler runtime R2015b (MCR v9.0). The amplitude for each frequency band was calculated through the FFT and the functional connectivity through the coherence method (https://neuroimage.usc.edu/brainstorm/ Retrieved on 13 April 2021) previously used [43]. Relative amplitude was calculated as a percentage for each frequency band from the total absolute amplitude spectrum (from 1 to 32 Hz). The localization of the abnormal activity source required the standardized low-resolution brain electromagnetic tomography (sLORETA) method [44].

### 2.3. Statistical Analysis

The calculation of the mean amplitude and coherence differences for each group were obtained using the Student’s T method. The 95% confidence intervals were estimated with a significance level of *p* < 0.05. The Bonferroni method was used to minimize errors arising from multiple comparisons.

Although we calculated several indices of functional association between pairs of electrodes that obtained good discriminative capacity, only significant ones are reported here. Their calculation was based on the sum of the coherences between the temporary locations (T3 and T4) with the Fz [index of functional discrimination between healthy and FM = (T3 − Fz coherence) + (T4 − Fz coherence)].

To elucidate whether the connectivity pattern may serve to identify patients with fibromyalgia, we calculated ROC curves through the SPSS ROC curve calculation, obtaining sensitivity and specificity values and a total discrimination index. This index was used alongside the clinical diagnosis to calculate the sensitivity/susceptibility, S = TP/(TP + FN), and the specificity, E = TN/(FP + TN), of the EEG as a diagnostic tool, where TP means “true positive”, TN “true negative”, FN “false negative”, and FP “False positive”. ROC curves were used as an accuracy index to explore the discriminative validity of EEG parameters. Note that no machine learning methods were used. The discriminative capacity, following the ROC curve analysis, using 95% confidence intervals (CI), was calculated using the following formula: sensitivity − (1 − specificity) [45]. The software SPSS v.23 was used for statistical analysis.

## 3. Results

### 3.1. Frequency Analysis

The FM group had significantly lower amplitude values than the control group (*p* < 0.001). These differences appeared in all frequency bands and locations studied, except for the relative frequency of the Delta band (absolute and relative frequency values appear in Figure 2 and the statistical figures Table 1).

Analyzing each group, frequency maps of the FM sample showed greater activity in the right parietal region (location P4) for the other bands. This seems to agree with the presence of spike-type grapho-elements in the right parietal and occipital areas, mentioned earlier.

### 3.2. Sources by LORETAs

All EEG registries were visually checked. A higher frequency of spike-type grapho-elements was found in patients (17.8%) than in controls (0%), with potential symptomatic seizures from irritation of neighboring cerebral cortical tissue located on the right occipital and parietal regions [35]. This is a common finding, as nonepileptic seizures (PNES) are frequently found in FM patients [46,47,48].

By analyzing the location of sources using the sLORETA method we located anomalous activity (signs of irritation) on the bilateral precuneus, with right predominance, on the right inferior parietal cortex, bilateral prefrontal medial cortex, and right anterior Cingular cortex for patients with FM. Figure 3 shows the localization of the band activity.

### 3.3. Analysis of Coherence

In the FM group, we found that cortical interconnections in FM were very scarce during eyes-closed resting, especially for Delta and Beta frequencies. Those existing in the Alpha and Theta bands were only visible in frontotemporal regions (*p* < 0.001). These findings contrast strongly with the degree of interconnection shown by control subjects. In particular, the coherence analysis revealed greater functional connectivity between the insular regions and the frontal regions in the patients with FM, but not in the control sample (*p* < 0.001). The coherence measures for FM and control samples appear in Figure 4, where only coherences equal to or greater than 0.5 were included.

### 3.4. Discriminatory Index

Finally, to distinguish patients with FM from controls, we analyzed a discrimination index using average amplitudes, the region of interest amplitude of P4, and functional connectivity between fronto-bitemporal locations. Statistical data appear in Table 2, showing good discrimination results with high accuracy (between 91.3–100%), and in Table 3, showing the area under the curve for ROC curves from frontal and temporal coherence values. The Theta band showed the best AUC with a sensitivity of cases of 100% and inclusion of 0% false cases. Frontotemporal functional connectivity showed a sensitivity of 91.3% and inclusion of 21.4% of false cases.

## 4. Discussion

In this study, we report the EEG data of a sample of 23 patients with FM and 23 matched controls to identify electrical differences between the two groups. The differential pattern of coherence, in the last instance, might work as a complementary method for FM diagnosis.

According to our results, patients with FM presented lower values than controls in all frequencies except for the Delta band. The frequency maps of the FM group indicated greater activity in parietal areas than in the rest of the scalp. Subsequent sources analysis indicated anomalous activity at the bilateral precuneus, with right predominance, on the right inferior parietal cortex, bilateral prefrontal medial cortex, and right anterior cingular cortex in the FM group. The coherence analysis of the brain signal showed clear differences between the groups, particularly in the bilateral frontotemporal region. Discriminatory analysis indicated a significant difference in interconnectivity patterns between patients and controls.

In general, we can state that frequency power was dramatically lower in the FM group than in the controls. These findings are compatible with the morphological findings reported by multiple authors using neuroimaging techniques [49]. The weight of the encephalon can be up to 3.3 times lower in chronic pain patients than in healthy subjects. In particular, this neuronal loss affects white matter, as evaluated by fractional anisotropy techniques [50]. With regard to location, studies by Kuchinad et al. (2007) showed that the most affected structures were the thalamus, medial posterior cingulate cortex, insula, prefrontal cortex, parahippocampus, hippocampus, anterior cingulate cortex, and striated nuclei [51]. A neuronal loss might explain the decrease in neuronal working synchrony, as a consequence of the communication defects caused by both the diffuse neuronal loss and the fibers of the white matter [52].

Deeper analysis of the affected bands shows that the most severe decrease took place in Alpha, Theta, and Beta frequency bands and was not significant in the Delta band. On the one hand, given that the first three bands depend on the cortical neuronal interaction and its networks, it is reasonable to link these alterations with the morphological impairments described above. On the other hand, Delta activity is more dependent on the somas of the deep pyramidal neurons affected in degeneration, usually by irrigation defects [53]. Other studies have reported similar differences between patients with FM and healthy controls while solving problems of various types. In global field power analysis, patients with FM presented lower modulation of Alpha and Theta, less synchronization, and lower spectral density, which indicates the presence of excessive neuronal noise [54]. According to fMRI research, patients with FM have to mobilize more cortical extension than healthy subjects, even in young patients aged between 25 and 40. This process may explain the presence of cognitive impairments. This reaction closely resembles the one in healthy elderly people, which is why some authors compare the consequences of fibromyalgia with an accelerated brain aging process.

Frequency mapping with a separate representation of the different frequency bands demonstrates that all bands showed their maximum amplitude in the right parieto-occipital region (Figure 5). This predominance may represent better conservation of the neuronal structure of the right parietal lobe than in other cortical areas (susceptible to potential irritative characters) [55]. This may also be related to the increase of glutamic acid and the decrease of gamma-aminobutyric acid described by Puiu et al. at the anterior cingular cortex, insula, and amygdaloid nucleus [56].

Analysis of the source with the sLORETA system pointed to the bilateral precuneus, right inferior parietal cortex, bilateral medial prefrontal cortex, and right anterior cingular cortex. These cortical structures are part of the default mode network, except for the insula, which was not observed in our maps. The most activated structure was the anterior cingular cortex, the first structure affected in FM, and the one in which has been observed the most manifest excess of glutamic acid, reduction in gamma-aminobutyric acid, and neuronal loss [24]. This fact may help explain why some authors argue that the exacerbation of pain in FM results from the existence of hypersensitive neural networks which, with their explosive response to any stimulus, cause synchronizations of the most sensitive networks [57]. These authors found a positive relationship between the intensity of pain caused by any normal stimulus and the degree of explosive synchronization of the EEG in these patients. According to the results of the current study, the presence of EEG spikes could be an example of these explosive phenomena, although it can also only facilitate them.

Concerning connectivity patterns, the intercommunication at the Alpha and Theta frequencies were more abundant in patients with FM, but short association fiber between neighboring areas predominate in the bilateral frontotemporal region, suggesting better conservation of U-fibers than long fibers in the whole of the white substance (Table 3). This contrasts with recent findings in which long fibers appeared more connected than short ones in the Alpha band [54]. Other studies support our results about the significantly lower level of connectivity signal in the FM group than in controls during resting state, particularly in the Theta band [27,58,59]. These functional connectivity differences are related to differences in pain intensity between FM patients and controls [60]. 

It also seems that some treatments that reduce pain in FM produce a change in brain connectivity, including the insula and the cingulate cortex, as in our study [61]. Physical exercise interventions also seem to normalize aberrant resting state functional connectivity in FM patients and to be associated with pain improvement [62]. These patterns of altered connectivity have been found to be associated with altered integration of sensory information [63]. Other studies of functional connectivity have shown a reduced pattern of connectivity in FM, and it has been suggested that functional connectivity could have clinical implications if used as an objective measure of pain dysregulation [64].

### Test Accuracy

The resulting patterns demonstrated high discriminability between patients and controls, predominantly in the Theta band and right frontotemporal regions (Table 3), agreeing with Ichesco [65]. A variety of methods of complementing FM diagnoses have reached high accuracy values: 72.9% accuracy of FM symptomatology questionnaires [66], 78.9% accuracy of neurophysiological reflex exploration of the spinal nociceptive flexion reflex, indicative of central sensitization [20], 95% accuracy of qEEG during polysomnography [67], 85.1% accuracy with different alternative criteria for FM (Salaffi et al., 2020), and 64.8%–71.3% accuracy based on physical examination and laboratory tests to identify FM patients [68]. Another study using an fMRI combination of neurologic pain signature, pain-related response, and multisensory nonpainful sensory stimulation showed 93% discrimination accuracy [29].

To our knowledge, EEG functional connectivity has not been used, so far, to discriminate between chronic pain syndromes. One future line of research should study the discriminative power of brain activity between different pain syndromes using functional connectivity, a line of research that needs further optimization. It is theoretically possible that chronic pain syndromes share common brain signatures, but discriminative methods could also be optimized to increase the accuracy of the results. For example, more specific examination with fMRI has been used to discriminate between pain syndromes with nonsignificant results, although 78.8% discriminative accuracies were achieved between FM and rheumatoid arthritis [69]. Overall, due to the consistently higher accuracy results of brain parameters over other methods of FM discrimination, neuroscientific methods could help improve FM diagnosis. In particular, EEG seems to be a promising tool with which to improve the early identification of patients at risk of developing a chronic pain condition.

Some possible limitations result from using the reference database, which is a relatively new (albeit increasingly adopted) procedure [70,71]. It is arguable that the FM sample could not be completely comparable with the nonclinical subjects extracted from the standard database because of incompatibilities between different acquisition methods. The use of several methodological techniques can, however, ensure the validity of the comparisons made in the present study, by reducing acquisition noise and sources of confusion [72]. These methods include the use of averaging mounts, band-pass frequency filters that exclude frequencies above 100 Hz [73], and relative power calculations to reduce differences individual measurements in skull thickness or amplifier calibration [74]. These techniques show similar results between acquisition devices when calculating the FFT bands [75], and similar signal-to-noise ratios between devices (up to 12 different ones). Even the comparison between modern low-cost EEG devices with medical-grade instruments shows comparable results for calculating frequency [76]. These techniques and new research opportunities are opening up new avenues for the use of open repositories in neuroscience research and collaboration. 

To confirm the validity of the control sample, other studies used the same database [10,77,78,79,80,81,82], including with FM population [83]. Alterations in functional connectivity are frequent in patients with FM (Hargrove et al., 2010), in particular alterations in frontotemporal connectivity [62,65,84], and are often associated with a lower white matter volume than in controls in frontal regions [28]. Our results replicate these previous findings. However, since we have specifically studied frontotemporal functional connectivity as a region of associations of interest (due to sample size limitations), these results should be viewed with caution. In summary, the comparison with normative samples constitutes a valid method if certain requirements are met (i.e., age-appropriate values, selection of artifact-free segments, and similar registry conditions, among others). Healthy control databases have been demonstrated to be reliable and to lack ethnic bias, making the comparison with clinical samples adequate, cost-effective, culture-fair, and highly sensitive to abnormalities in brain function both to spectra [15] and LORETA comparisons [85]. 

About the sample size, it would have been interesting to compare patients with FM with depressed individuals to see whether our algorithm correctly differentiates from other comorbid psychiatric pathologies. Another limitation is that there may be different origins of FM which end in the same syndrome. Therefore, we may be facing a particular type of FM pattern that presents specific connectivity abnormalities. However, the syndromic sub-groups of FM remain an unsolved problem. We have not been able to measure pain levels or precise psychopathological characteristics through questionnaires or cognitive tests, which may be an interesting line of work in the future, as suggested by some studies [86,87]. Triñanes et al. suggested that types I and II of FM differ by psychopathological profile [87].

This study shows potential distinctive neurophysiological features in patients with FM that put in connection function and structure. In the future, data could be the basis of a reproducible and safe diagnostic method for FM, as previously suggested by other studies [88].

## Figures and Tables

**Figure 1 jcm-10-03277-f001:**
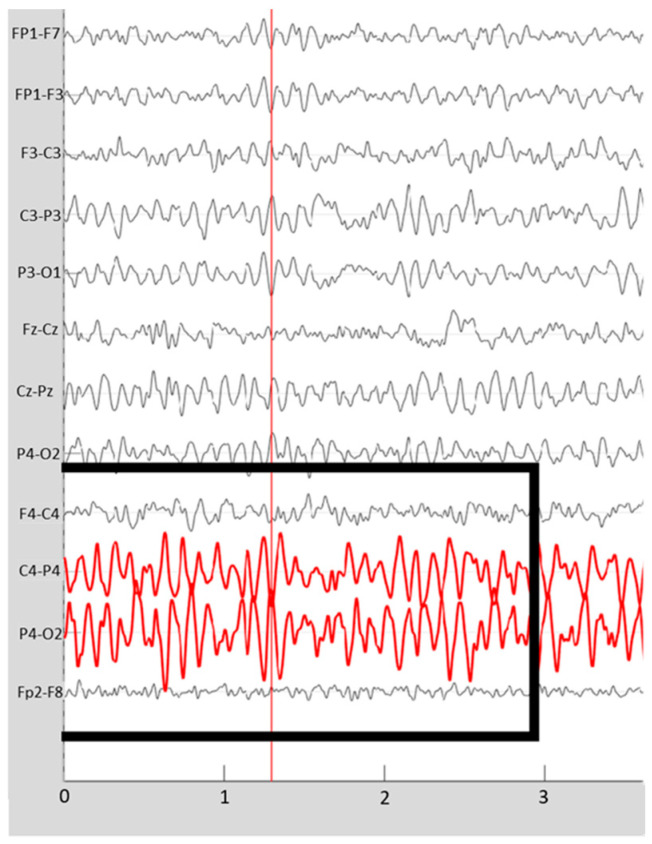
Example of EEG registry that presents abnormal activity (squared in black). Recording details: amplitude 70 microvolts, 10 s.

**Figure 2 jcm-10-03277-f002:**
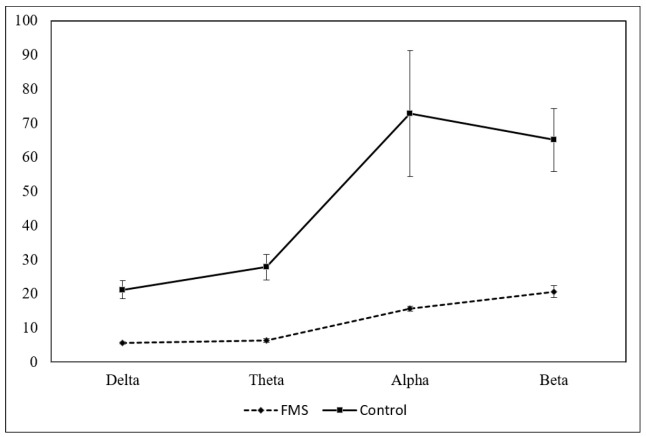
Absolute amplitude average of frequency bands (µV) in FMS and control groups (*p* < 0.0001 for all frequency bands comparisons).

**Figure 3 jcm-10-03277-f003:**
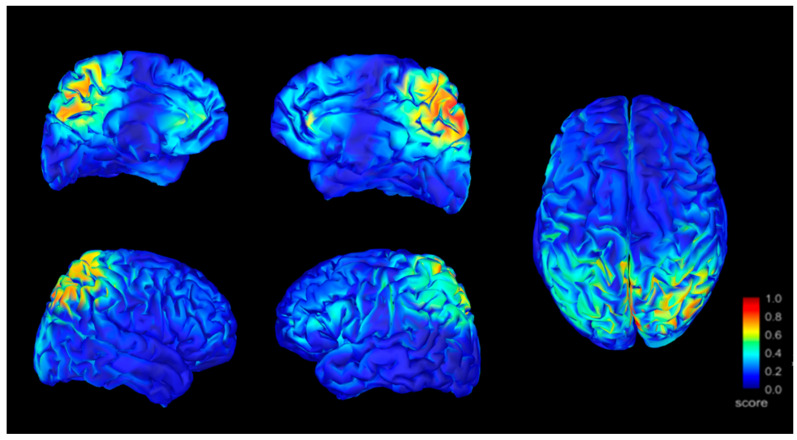
Source localization of abnormal activity in FMS patients.

**Figure 4 jcm-10-03277-f004:**
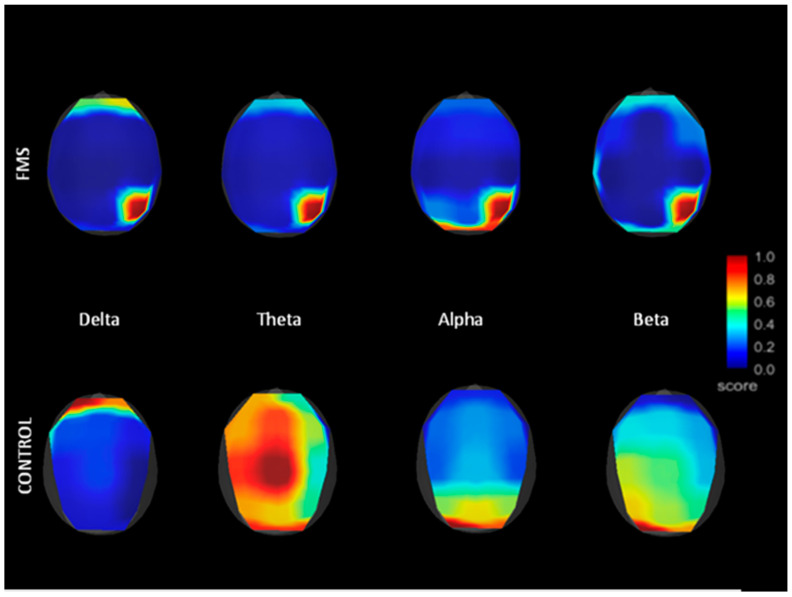
Grand average representation of FFT frequency bands for FMS and control subjects. Amplitude was reduced in FMS patients for all electrode positions.

**Figure 5 jcm-10-03277-f005:**
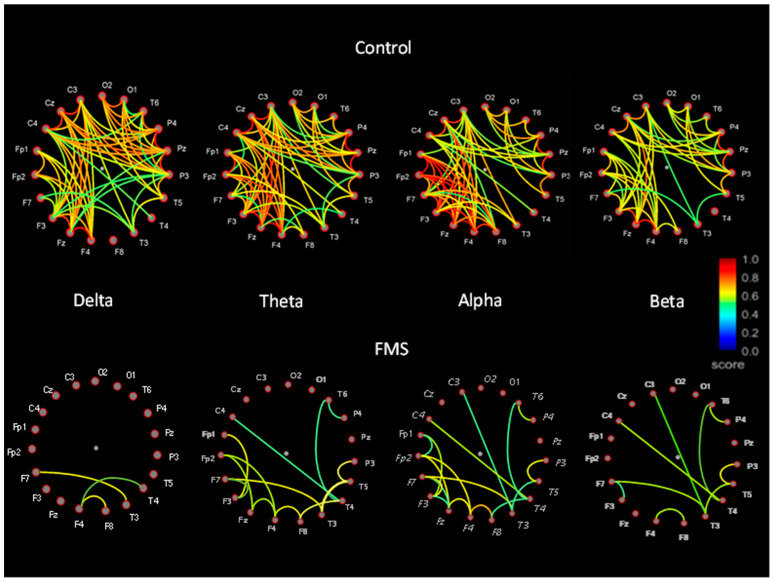
Functional connectivity comparison between control and FMS samples (connectivity between temporal and frontal locations).

**Table 1 jcm-10-03277-t001:** Differences in frequency bands between groups of absolute and relative mean frequencies for FM and control group * *p* < 0.001.

	FM (*n* = 23)		Control (*n* = 23)		
	Mean	SD	Mean	SD	*p*
ABSOLUTE					
Delta	5.59	0.42	21.22	2.69	0.000 *
Theta	6.32	0.50	27.88	3.75	0.000 *
Alpha	15.74	0.75	72.86	18.53	0.000 *
Beta	20.67	1.72	65.11	9.22	0.000*
RELATIVE					
Delta	11.57	0.49	11.57	2.07	0.991
Theta	13.08	0.65	15.02	1.55	0.000 *
Alpha	32.62	1.06	38.45	4.09	0.000 *
Beta	42.72	1.17	34.97	2.08	0.000 *

**Table 2 jcm-10-03277-t002:** Comparison between discriminative indexes for different EEG parameters. AUC = area under the curve.

FFT Amplitude	ROC (AUC)	Sensitivity/ 1 − Specificity	*p* [95% CI]
Delta (1–4 Hz)	0.618	0.643/0.609	0.234 (0.417, 0.819)
Theta (4–7 Hz)	1	1/0	0.000 (1, 1)
Alpha (7–14 Hz)	0.975	1/0.217	0.000 (1, 1)
Beta (15–32 Hz)	0.988	1/0.174	0.000 (0.960, 1)
Right Parieto-Occipital activity (P4)	0.984	1/0.217	0.000 (0.951, 1)
Frontotemporal functional connectivity	0.913	0.913/0.214	0.000 (0.816, 1)

**Table 3 jcm-10-03277-t003:** Accuracy index—area under the curve (AUC) for ROC curves calculated from coherence values. * *p* < 0.05; ** *p* < 0.001.

Derivations	Delta	Theta	Alpha	Beta
Fp1-Fp2	0.390	0.467	0.294	0.238
Fp2-T4	0.843 **	0.907 **	0.875 **	0.846 **
Fp1-T3	0.843 **	0.849 **	0.884 **	0.746 *
T3-T4	0.580	0.596	0.706	0.593
Fz-T3	0.712 *	0.684	0.780 *	0.765 *
Fz-T4	0.799 *	0.835 *	0.774 *	0.846 **
Fz-T3 + Fz-T4	0.877 **	0.794 *	0.788 *	0.822 *
Fp1-T3 + Fp2-T4	0.765 *	0.887 **	0.913 **	0.80 *

## Data Availability

Data from healthy controls were randomly collected from a public standard database available on the Internet under the auspices of the National Center for Research Resources of the National Institutes of Health.

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
