# Peer review of "Fibromyalgia Detection Based on EEG Connectivity Patterns"

_jcm, 2021, doi:10.3390/jcm10153277_

Round 1
Reviewer 1 Report
This study described the connectivity patterns in a group of patients with FM and found a pattern that fits with the clinical characteristics of the disease. The research was well conducted and the results were worthy of publication. However, there are some issues that needed clarification.
1. About the introduction and rationale of the study: why was EEG connectivity being investigated? Was it because the author expect a potential use of the EEG connectivity as a diagnostic tool should the results provide strong support? From a theoretical viewpoint and neurophysiological mechanism perspective, why would patients with FM display a certain pattern of EEG signals at resting state? and why such pattern would be different from controls? This physiological foundation needed to be addressed and provided in the introduction.
2. The control group appeared to have n=14 participants. This was wrongly worded in the abstract (line 18 page 1) and in the methods (line 65 page 2). Was there n=23 initially but because of drop out or other issues that remained n=14 for the control group? this needed to be corrected or clarified.
3. Authors mentioned a matched sample. How was the control group matched with the patient group? Matched in term of age? sex? or other parameters? The demographic data of the control group needed to be presented (it was currently missing). Also, because of unequal sample size between the two groups, the authors need to be explicit how these two samples are matched.
4. For the patient group, were there any neurological diseases or medical history that would interfere with EEG signals, say epilepsy? A statement is needed to clarify, even if none.
5. On page 6, lines 204 to 206 (figure 4), the interpretation of the figure was that "maximum amplitude was found in the right parieto-occipital region". However, on figure 4, those connections with red line (higher beta) were clustered within connections between the frontal regions and frontal-central regions, with a few connections between parietal and central regions and most parietal and occipital connections are of lower beta. This lead to an interpretation that maximum amplitude of connection was instead between the frontal especially for theta and alpha wave bands. Would the author please clarify these results and interpretation.
6. Would there be any relationship of EEG connectivity with the history/duration of FM? About 77% of patients have over 12 years of FM, how about the rest? Would this factor contribute to variation of EEG connectivity patterns?
7. Are there any limitation to the study?
8. On page 7, last paragraph, the sentence was incomplete or structurally wrong for the first sentence, making it hard to understand (lines 239 to 240), Maybe just keeping the first part of the sentence, i.e., "This study shows potential distinctive neurophysiological features in patients with FM". After that, begin a new sentence to communicate the characteristics of functional connection found in the study.
9. Also the last paragraph on page 7 (line 241 to 242), the authors mentioned that the data could be at the basis of reproducible and safe diagnostic method. This would require some elaboration as to how such findings lead to diagnostic application.
Author Response
Dear Reviewer 1,
Many thanks for your comments. We have worked on the responses and modified the content on the second version of the manuscript. Please, find bellow our comments.
REVISOR 1:
This study described the connectivity patterns in a group of patients with FM and found a pattern that fits with the clinical characteristics of the disease. The research was well conducted and the results were worthy of publication. However, there are some issues that needed clarification.
- About the introduction and rationale of the study: why was EEG connectivity being investigated? Was it because the author expect a potential use of the EEG connectivity as a diagnostic tool should the results provide strong support? From a theoretical viewpoint and neurophysiological mechanism perspective, why would patients with FM display a certain pattern of EEG signals at resting state? and why such pattern would be different from controls? This physiological foundation needed to be addressed and provided in the introduction.
RESPONSE:
The study of connectivity is increasingly relevant, being able to identify specific malfunctioning areas. In the case of FM, diagnostic tests (MRI, fMRI and EEG, among others) can be used to detect these abnormalities. The new version of the manuscript includes several references regarding the connectivity impairments identified in FM patients, both with fMRI and EEG methods (see below). However, EEG is a non-invasive, accessible and transportable method, so its performance speeds up the diagnostic process largely.
The new version of the test states (lines 72-87),
“Based on these data, some observations confirm that functional connectivity is generally impaired in FM patients (7-9). In particular, recent FM electroencephalography (EEG) studies during the resting state showed abnormalities in Theta (10) oscillation and a decrease in the power spectrum in the 9-14 Hz band in the central regions compared to controls (7). In agreement, another study found reduced lower frequencies and increased higher frequency activities in FM patients with respect to controls (8). Even other neuroimaging techniques, such as functional magnetic resonance imaging (fMRI), showed disrupted connectivity at the Theta frequency for FM patients compared to controls in de-fault network (9) and resting-state sequences (10). However, there is so little information about the electrical coherence between brain regions in FM patients compared to controls (11) and specific connections between regions aren´t addressed yet. The analysis of such differences might be based on a diagnostic method for FM using the EEG technique (12). In this research, we attempt to identify the electrical connectivity differences between patients with FM and matched control groups using EEG analysis and, specifically, studying the differences in functional connectivity patterns.”
Also, there are many studies of functional connectivity by EEG in the psychiatric and neurological population. The reason is to replicate those findings in patients with fibromyalgia. From the outset, the loss of cognitive function in FM, pain symptoms and the anhedonic symptom (González-Roldan et al., 2013, Napadow et al., 2013, Ceko et al., 2013). To clarify this aspect, the manuscript has been modified (line 87-89) as follows,
“The hypothesis is that patients with fibromyalgia present significant differences in EEG activity concerning controls and that this differential activity would be linked to a decrease in brain connectivity, as it happens in other diseases (13–15).”
- The control group appeared to have n=14 participants. This was wrongly worded in the abstract (line 18 page 1) and in the methods (line 65 page 2). Was there n=23 initially but because of drop out or other issues that remained n=14 for the control group? this needed to be corrected or clarified.
RESPONSE:
We are deeply sorry about that typo. The correct figures are 23 patients and 23 controls.
This information is now corrected at the new version of the manuscript.
- Authors mentioned a matched sample. How was the control group matched with the patient group? Matched in term of age? sex? or other parameters? The demographic data of the control group needed to be presented (it was currently missing). Also, because of unequal sample size between the two groups, the authors need to be explicit how these two samples are matched.
RESPONSE:
Demographic data is now presented, where sex and gender are presented for both samples. Since the sample size typo is now corrected, the manuscript present equal sample size between the two groups. The match was done base on sex and age. In agreement, the text now states (Lines 100-108),
“The experimental group was 56 years old on average (range 35-65). From them, 77% had more than 12 years of evolution of the disease. For the age-gender-matched control group, 23 healthy controls were randomly collected from a public standard database available on the Internet under the auspices of the National Center for Research Resources of the National Institutes of Health (16). This procedure of comparing Fibromyalgia EEG data with non-clinical population data from public repositories has been previously published with similar results (22). Gender distribution was of five males (21.8%) and eighteen females (78.2%) for both groups. T-test analysis indicated there were no significant differences in age (t (44)=.061, p=0.54).”
- For the patient group, were there any neurological diseases or medical history that would interfere with EEG signals, say epilepsy? A statement is needed to clarify, even if none.
RESPONSE:
Control group participants inclusion criteria was to be free of medical conditions. The current version of the manuscript now includes (Line 133),
“No pathological signals were found on the EEG registry of the control sample”
- On page 6, lines 204 to 206 (figure 4), the interpretation of the figure was that "maximum amplitude was found in the right parieto-occipital region". However, on figure 4, those connections with red line (higher beta) were clustered within connections between the frontal regions and frontal-central regions, with a few connections between parietal and central regions and most parietal and occipital connections are of lower beta. This lead to an interpretation that maximum amplitude of connection was instead between the frontal especially for theta and alpha wave bands. Would the author please clarify these results and interpretation.
RESPONSE:
The electrical amplitude (microvolts, extracted from the FFT) can be mistakenly identified as the intensity of the connectivity (which would be as corrected correlation, without amplitude unit). In reality, in healthy conditions, there is less connectivity when there is more amplitude. (It can be explained by the presence of right parietal foci, which increases the inhibitory barrier and therefore reduces connectivity with other regions, leaving the central-frontal regions more interconnected than the parietal regions). This occurs in other painful conditions such as chronic jaw pain (https://www.sciencedirect.com/science/article/pii/S2213158219303146).
This may also be related to a worsening of cognitive function, processing speed, and perhaps a reduction in the flow of information reaching the frontal-frontal network. What is also interesting is that there is a general reduction of the front intercom, leaving only local connections (C3-T3; T3-T5; P3-T5; F7-F3 and the same for the H.D.). This can be also found in other cases (https://jnnp.bmj.com/content/82/5/505; https://onlinelibrary.wiley.com/doi/full/10.1111/epi.12057)
In any case, this data is consistent with a lower interconnectivity of the subsequent networks and the preservation of the front networks in FMS as can be seen in this article https://www.ncbi.nlm.nih.gov/pmc/articles/ PMC5484465; (https://www.sciencedirect.com/science/article/pii/S2213158219303146).
- Would there be any relationship of EEG connectivity with the history/duration of FM? About 77% of patients have over 12 years of FM, how about the rest? Would this factor contribute to variation of EEG connectivity patterns?
RESPONSE:
The size of the sample (n = 23) does not allow us to establish valid correlations taking the years since diagnosis as a variable. However, we can compare it with other studies such as Gonzalez-Roldán et al, 2016, in which an effect of the history of disease was found in the patterns of connectivity of patients with FM compared to healthy controls.
- Are there any limitation to the study?
RESPONSE:
The new version of the manuscript includes limitations regarding design and sample size and further issues (lines 370-379), as follows,
“In relation with the sample size, it would have been interesting to compare FM patients with depressed individuals to see whether our algorithm properly differentiates from other comorbid psychiatric pathologies. Another limitation is that there may be different origins of FM that end in the same syndrome. Therefore, we may be facing a particular type of FM pattern that presents specific connectivity abnormalities. However, the syndromic sub-groups of FM remain an unsolved problem. We have not been able to measure pain levels, or precise psychopathological characteristics through questionnaires or cognitive tests, which may be an interesting line of work in the future as suggested by some studies (49,50). Actually, Triñanes et al suggested that types I and II of FM differ by psychopathological profile.”
- On page 7, last paragraph, the sentence was incomplete or structurally wrong for the first sentence, making it hard to understand (lines 239 to 240). Maybe just keeping the first part of the sentence, i.e., "This study shows potential distinctive neurophysiological features in patients with FM". After that, begin a new sentence to communicate the characteristics of functional connection found in the study.
RESPONSE:
We have rewritten the phrase to make it easier to understand (lines 380-382), as follows,
“This study shows potential distinctive neurophysiological features in patients with FM than put in connection function and structure.”
- Also the last paragraph on page 7 (line 241 to 242), the authors mentioned that the data could be at the basis of reproducible and safe diagnostic method. This would require some elaboration as to how such findings lead to diagnostic application.
RESPONSE:
Undoubtly, from this initial experiments to the use of this idea into the clinical practice would be necessary to run many more tests and analysis. However, finding key fifferential points between healthy patients and control is essential to progress into the FM diagnosis. Being so, the last sentence was modified in the currerent version of the manuscript (lines 352-353), as follows,
“In a latter step, data could be at the basis of a reproducible and safe diagnostic method for FM, as previously suggested by other studies (51).”

Reviewer 2 Report
MANUSCRIPT REVISION
“FIBROMYALGIA DIAGNOSIS BASED ON EEG CONNECTIVITY PATTERNS”
GENERAL COMMENTS
The manuscript refers to a hot and highly interesting topic on a disease hardly needing accurate and reliable diagnostic tools. With this respect, the study aim is valuable. The authors have performed a case-control study on fibromyalgia patients and controls, assessing resting-state EEG patterns using frequency analysis with a special focus on functional connectivity. Their results disclose differences between studied populations.
However, I have serious issues both at a conceptual level (see below details) and in the way the manuscript is written (language).
I gave my remarks following page and section orders. Thus, although I did not mention pages etc, it should be easy to find back which manuscript part I am referring to.
MAJOR REMARKS
Research question and working hypotheses
Although the research question might look evident, it is not clearly formulated. The term diagnosis should be further explained. In fact, there is no conclusive and unequivocal clinical diagnostic criterium for fibromyalgia; rather a constellation of symptoms that have been defined by experts, with all limitations the authors evoked. Thus, from a clinical perspective, a diagnostic tool should either be complementary to clinical criteria (thereby bringing more specificity and/or sensitivity to them), specifically correlate to one or a few highly evocative symptoms (hence strengthening their role in the diagnosis) or be itself sensitive and specific enough to diagnose fibromyalgia (and become the “goldstandard” diagnostic tool).
It appears as if the authors relate their research question to cognitive defect in fibromyalgia, however this is not the most specific/evocative impairment in fibromyalgia. So, if the authors mean EEG patterns they are seeking for would be related to cognitive deficit, they should clearly state it, and say why these patterns should be considered to be specific to fibromyalgia in absence of specific cognitive impairment reported in this disease. I would expect the authors to talk about pain. They rather evoke hyperalgesia and allodynia as the key elements resulting from impaired brain excitability, while in the current understanding, it is difficult determining exactly which symptoms brain hyperexcitability is related to. In general, chronic pain is taken as a whole in this context, with accompanying symptoms such as hyperalgesia or allodynia. Anyhow, none of these symptoms is described in studied patients; therefore, no association with EEG patterns can be assessed.
Thus, the authors should solely focus on EEG patterns when stating their research question, while making association with clinical factors would be of secondary relevance. With regard to this, when stating their research question in the manuscript (introduction), they do not say how EEG patterns could diagnose or contribute to diagnose fibromyalgia. They should further elaborate, based on existing literature and from their previous data (if any), on how differences in power spectra, coherence, etc would validate their diagnosis concept of fibromyalgia (here solely based on EEG data); which frequencies they target and why; in which brain areas they expect differences and why so etc. Otherwise, it would appear as if authors go “fishing” for differences between patients and controls and build thereafter hypotheses about the meaning of observed differences.
To summarize, the whole introduction should be re-written with clear research questions, hypotheses etc. The authors cannot only say “recent findings”, “showed abnormalities” etc. The manuscript would gain in clarity if the literature supporting stated hypotheses was further detailed.
Study design and procedure
Selection of fibromyalgia patients was well-conducted. However, it would have been interesting to also have their demographic and clinical characteristics (especially related to pain and other typical dysfunctions) in order to estimate the severity of the disease (and possibly improve result interpretation even this was apparently not the primary aim of the study). It is not clear from the methods section whether the controls had already been submitted to EEG recording in another context or were selected from existing repositories and thereafter EEG-recorded in the frame of this study. Instead of justifying why doing what they did constitutes a valid method, the authors should clearly specify how EEG data from controls were acquired and show that the used methods allowed reliable comparison with data from fibromyalgia patients. Otherwise, comparing their patients’data with a kind of a “black-box” database raise important methodological questions that need to be specifically addressed in order to validate the present study design.
Interpretation of EEG data is very questionable. It is not clear whether the authors are talking about pathological EEG patterns (e.g. sharp waves) which refer to other clinical entities such as epilepsy; and this is far away from fibromyalgia and cannot be claimed for as a diagnostic tool (or should be clearly demonstrated); or they are seeking for patterns that are not known to be pathological in other clinical contexts, but are prominent enough to be relevant for the present analysis. In both cases, the authors should more clearly justify their a priori interpretation frame and support it with existing literature and/or strongly motivated assumptions.
Concerning quantitative EEG measures, authors evaluated mainly different frequencies bands, their specific amplitudes, and functional connectivity. However, it is not clear to me why they compiled several indices and variables and how the latter contribute to support their hypotheses and to fibromyalgia diagnosis. The authors should clearly motivate why they used these variables and all irrelevant ones removed from this analysis.
The method section concerning EEG data lacks important technical information and should clearly specify in a systematic way the procedure for: 1) data recordings (the recording sampling rate is lacking); 2) data preprocessing (artefact removal, filtering, down-sampling if any etc); 3) data analysis according to different techniques and their respective outputs in relation with stated hypotheses and the three above-mentioned parameters (frequencies, amplitudes and connectivity).
It is not immediately clear which frequency power the authors analyzed (in the sensor space or source-related); this should be further explained. On the other hand, I do not understand why authors analyzed connectivity and power related to individual electrodes (or montage) instead of functional (or anatomical) brain areas. Probably some mapping using reference Atlas would be useful here? But this is not my specific domain of expertise (though, I think this should be well explained to lay readers).
Statistical analysis should be directly related to operationalized hypotheses and purposely explained to the reader. The methods used should be validated elsewhere or proactively justified (no reference is provided for statistical analysis). Could the author explain what they mean by sensitivity and specificity of their analysis in this context? Statistical analysis does not itself provide these validations; there should be a clinical (or at least conceptual) thought behind and this is lacking in the explanation given for this ROC curves analysis. If the authors cannot provide one clearly stated conceptual interpretation frame for this data analysis, these results should be left out.
Otherwise, it looks like many of performed analysis do not bring any added value to the manuscript. They should therefore be removed.
As mentioned above, data related spike analysis, unless clearly associated to fibromyalgia and to stated hypotheses, should be removed from the manuscript. I do not see their relevance in the frame of this kind of study.
Results
The result section should follow the frame of above-mentioned hypotheses, data collection and analysis and any data not related to them removed from the manuscript. Thus, the main data should present in a systematic way: 1) frequency power in different bands, 2) Source analysis, 3) coherence. All other data should be either related to these three main sections or removed from the manuscript.
Discussion
The discussion should remain focused on the research question and on the hypotheses. Likewise, data interpretation should evoke for example only clinical factors related to EEG patterns that were disclosed by data analysis. The main discussion seems to be conducted on spatial localization and anatomical determinants, while more electrophysiological, functional discussion is left away and the way different findings are related to the diagnosis of fibromyalgia is not demonstrated.
I cannot further comment the discussion before it is adapted to corrections made in other sections.
MINOR AND DETAILED REMARKS
Abstract
Either structured or not (according to the Journal guidelines), the abstract should summarize the research question, working hypotheses and the study aim first. The methods section suggests that controls were recorded in the frame of this study while this is not clearly said in the manuscript. Please say what was really done. Show only data and analysis that are relevant for the study. The conclusion should state in which way observed neurophysiological patterns contribute to the diagnosis of fibromyalgia and be in accordance to what is said in the manuscript. The patterns “fitting with clinical features” was not really studied, neither specifically discussed… It appears as overinterpreting obtained data. The authors should remain in the frame of what they studied and analyzed.
Several clinical terms and other concepts are used in a confusing way that is not always correct. For illustration purpose, I will give some of them below, but the whole manuscript should be reviewed with regard to that. The English language need thorough editing. The way the manuscript is written, it seems that the authors used automated web-based translators. It gives a bad impression reading uncomplete and non-structured sentences or with what appears to be a literal (most of the time inappropriate) translation.
Concepts
Introduction
Fibromyalgia is not characterized by “pain and allodynia”, but by pain and accompanying features such as allodynia, hyperalgesia etc. What are “associated neurological symptoms of central origin”? to my knowledge, except some sensory disturbance, there are no prominent neurological deficit and even if there could be some, their “central” origin is not documented. Further, the authors say later “the current literature does not report peripheral or central functional impairment justify such symptoms”, which is contradictory to what they just said before.
What do authors mean by cognitive symptoms being “independent” symptoms in the frame of fibromyalgia?
The definition of hyperalgesia and allodynia is not correct and the way the authors explain the origin of hyperalgesia is not is accordance with the most accepted litterature. What is “efferent antalgic system”? What is excessive “cognitive-affective influence in pain experience”? These concepts should be well clarified and defined according the general understanding in the field and related to the study. I do not find the relation the authors establish between these statements and impaired functional connectivity. This has to be clarified. Abnormal EEG patterns should be detailed because this is the basis of the study… Just saying “impairments” or abnormalities gives no relevant information. The authors should further motivate how “the analysis of EEG differences” should constitute a basis for fibromyalgia diagnosis, because this is the key research question.
Material and methods
The mean age of controls should be mentioned. The patients’medical history should be given to allow interpretation of EEG spikes (and exclude epilepsy).
The sentence concerning the informed and signed consent is not understandable.
The age of the MD who visually analyzed EEG data is not relevant for the study. Are posterior sharp-wave the only pathological patterns observed in EEG? This is strange! I would expect some slowing potentially related to cognitive impairments (if any).
Abbreviations should be entirely written when first mentioned in the manuscript (e.g. FFT).
Results
The authors when talking about frequency band, mention spikes…. I think spikes should be differentiated from background activity and be separately analyzed. Anyhow, the interpretation in the result section refers to potential “symptomatic” epileptic seizures. This is really problematic! First, epileptic seizures are not, to my best knowledge, part of fibromyalgia symptoms and cannot therefore not be considered as contributive to the diagnosis. Second, in order to say “symptomatic” one has to to mention clinical symptoms related to abnormal electrical activity, which is not the case in this study. Thus, once more, although observation of spikes in fibromyalgia patients could be relevant in some respect, it has nothing to do with the present study. It should probably be mentioned or studied elsewhere, with appropriate clinical investigations in complement and be discarded from this manuscript.
Absolute and relative spectral powers are indifferently presented in the text. They should separately be presented, unless similar (which does not appear to be the case here).
Using electrode naming for brain area localization appears quite uncertain to me, knowing the poor localizing value of EEG. Please use reference Atlas to confirm these localizations. What is the relevance of showing sensor space localization and right after the source localization? Which difference do the authors make between the two approaches? Please explain!!
In relation with right-lateralization, could the authors specify the manual handedness of participants? Is there any lateralization in the control group? How is this finding related to the study questions?
The sLORETA apparently analyzed localization of spike source. Is this an issue for this study? Please refer to my remark above concerning the use of spikes in this study.
I do not understand at all what is the discriminatory and/or accuracy index and most of all what it/they bring(s) to the study. Why did the authors perform this analysis?
Tables and figures are poorly labeled/explained. Figure legends lack. The graph on figure two has no ordinate units.
Discussion
Should be totally written again. I will not comment in detail, but the use of terms is once more not appropriate and highly confusing.
E.g.:
- frequency band value means nothing! It is the frequency power (please state whether in the sensor space or source-related)
- “encephalic mass”
- neuronal loss affecting white matter and thereafter, mention of exclusively grey matter areas
- neuronal synchrony is primarily a functional concept; should first be discussed in this perspective, before jumping into neuronal loss (that was neither studied or indicated in collected data).
- what is irrigation defect??? Please explain
Authors mention studies reporting “differences between patients and controls while solving problems of various types” is this during experimental tasks? Is this in the daily life activities? And what is the link with the study findings? Authors appear to explain mechanisms leading to cognitive impairments. This should be more appropriately discussed and the link with the study results more directly shown.
The authors interpret the right-lateralized maximal power in the parietal lobe as related to better conserved neuronal structure in this region compared to other cortical areas susceptible to irritative activity, while they found the source of spikes in the same right parietal area (see sLORETA results). This interpretation appear to be contradictory. Relationship with increased glutamatergic and decreased GABAergic signaling is not understandable the way explained… Once more, I do not get the point.
The interpretation of sLORETA data in relation with spikes etc is not relevant to me (at least the way presented and explained). Same for the connectivity patterns. I wonder what electrode connectivity means conceptually?? Since this is not my domain of expertise, I will not further comment.
CONCLUDING REMARKS
Overall, although some data obtained in the frame of this study might be of some relevance to the understanding of fibromyalgia mechanisms and characterization, I feel the authors to some extent performed a data-driven study conception and analysis. To my view, the authors missed data interpretation in relation with the research question they targeted and even to some extent overinterpreted their data.
The whole research question, working hypotheses, data analysis should be conceptualized again and reframed. Serious methodological issues should be solved. Thereafter, appropriate data interpretation made strictly in the limits of what was studied and obtained results. The manuscript must be fully written back accordingly and language editing performed.
Author Response
Please, see the attached document.

Reviewer 3 Report
I was initially excited by the prospect of this manuscript which aimed to evaluate how resting state scalp EEG could vary between patients with fibromyalgia and healthy controls. Unfortunately, as described above, there are structural methodological issues with the experimental design that cannot be fixed easily without new data acquisition, and entirely invalidate the results of the experiment. Based on these structural issues, I did not comment in detail about the interpretation of the results.
Major Comments
- The EEG information from patients with FM was acquired as part of the study, whereas the healthy controls were acquired as part of a standard database with different technology, location, and acquisition parameters. Unfortunately, EEG is sensitive to acquisition parameters and functional connectivity is an additional detailed analysis of EEG data, which may be entirely marred by difference in acquisition details. Due to this critical confound, it is impossible to determine if the observed results are due to these differences in acquisition or associated with fibromyalgia. The concern for this confound is raised based on the marked difference in absolute power in spectral bands as compared to the relative power in each spectral band.
- The authors state that characterization of objective peripheral or central neurological structural deficits are needed to define fibromyalgia. However, fibromyalgia is a clinical syndrome that likely reflects a change in the function as compared to the structure of the nervous system. This is reflected by fibromyalgia being defined by rheumatology as compared to neurology.
- Due to concern for multiple testing, please at least list the pairs of electrodes that were evaluated prior to showing the electrodes with significant findings.
- The authors state in the abstract that they performed ROC analysis. The values from the ROC evaluate the predictions from a machine learning prediction algorithm and the ROC is not a method in itself. The authors should specify their machine learning methods in the abstract. In the main text, it seems like they only report a single amalgam feature as compared to developing a machine learning model.
- Due to the likelihood for evaluating many potential amalgam features and only reporting one, with no discussion of cross-validation or training/testing groups, I have high concern for overfitting to this specific small dataset, and that these results will not generalize to a broader population of patients with fibromyalgia.
- The authors should provide a citation for the direct and indirect expenditures associated with FM when they discuss it in the introduction.
- The author’s initial hypothesis of “lower” EEG activity is quite vague and nonspecific. There is not a meaningful biological hypothesis for why the patient may have “lower” EEG activity. The definition of global lower activity also is unclear, as attenuation of EEG voltage can be seen in many severe conditions including global hypoxic-ischemic damage and bilateral subdural or subgaleal injuries. This is differentiated from other changes in EEG caused by global cerebral dysfunction like encephalopathy, where the predominant frequency of EEG shifts slower, so there’s global slowing of the EEG.
- In the procedure section, the authors state that “patients usually show the presence of areas with atrophy or signs of aging or irrigation alterations.” Based on my clinical experience and knowledge of the neuroimaging research in fibromyalgia, typical imaging findings for patients with fibromyalgia are normal. Therefore, the authors must provide a citation to support their claim.
- In the procedure, please provide a citation of the coherence method for analyzing functional connectivity.
- From Table 1, the authors report power in each spectral band, but they do not specify the location of the channels where power was calculated. Was this a weighted average across all channels with an average reference, or another combination of channels?
- The authors state that the EEGs were visually evaluated and showed 17.8% spike type grapho-elements in patients with fibromyalgia. Please specify the expertise of the reader, as EEG reading is difficult and requires many years of clinical training. Based on the term used, it’s difficult to know if these spikes are artifact or suggestive of other pathology, because the term used does not conform to the standards of the Clinical Neurophysiology Society.
- The authors should show a tracing of the anomalous activity on EEG, because it’s only present in 18% of patients and it is unclear how to interpret it.
- The number of matched controls is inconsistent between the methods and discussion. Please clarify this to ensure accuracy. There’s a marked difference between utilization of these methods with unbalanced groups (14 controls) and balanced groups (23 controls).
Minor comments:
- Spectral power is a logarithmic quantity, so the author should perform student’s t-tests on the log of spectral power in each frequency band, as compared to the absolute value.
- When the authors reference fibrofog, they should include a citation on the topic at the end of the sentence.
- The tone of the introduction in the abstract and main text are more extreme than typically seen in formal writing about clinical conditions.
- In the introduction, the authors should define the “Alpha-2” band as alpha has a standard definition but “Alpha-2” does not.
Author Response
Dear Reviewer 2,
Many thanks for your comments. We have worked on the responses and modified the content on the second version of the manuscript. Please, find below our specific responses.
I was initially excited by the prospect of this manuscript, which aimed to evaluate how resting state scalp EEG could vary between patients with fibromyalgia and healthy controls. Unfortunately, as described above, there are structural methodological issues with the experimental design that cannot be fixed easily without new data acquisition, and entirely invalidate the results of the experiment. Based on these structural issues, I did not comment in detail about the interpretation of the results.
Major Comments
- The EEG information from patients with FM was acquired as part of the study, whereas the healthy controls were acquired as part of a standard database with different technology, location, and acquisition parameters. Unfortunately, EEG is sensitive to acquisition parameters and functional connectivity is an additional detailed analysis of EEG data, which may be entirely marred by difference in acquisition details. Due to this critical confound, it is impossible to determine if the observed results are due to these differences in acquisition or associated with fibromyalgia. The concern for this confound is raised based on the marked difference in absolute power in spectral bands as compared to the relative power in each spectral band.
RESPONSE:
The reviewer indicates his/her worries about the possibility of confounding effects derived from the use of standard database for the control group. He/she also indicates that his/her concerns were raised by the differences in absolute and relative amplitudes. We thank the reviewer for the opportunity to clarify this to the reader. We have added a comment on this in the limitation sections explaining why we think that large differences found between FMS and controls cannot be attributed to acquisition parameters and must be interpreted as biological sources. Many control measures were applied to reduce the introduction of confound sources of the acquisition procedure to assure that differences are due to biological differences between FM and controls.
First, differences should not be attributed to device differences, because there is evidence that even different technologies in devices (up to 12 devices showed similar signal to noise ratio) do not show differences when calculating FFT bands, suggesting that acquisition quality might not be of primary importance when calculating FFT bands in contrast to ERP acquisition (Bussalb et al., 2018). Furthermore, the electrical amplitude of modern EEG amplifiers are very similar and comparable even in the cheapest ones (Barham et al., 2017; Frey, 2016).
Second, average reference was applied to both samples, which is a good method to reduce the noise to EEG signal ratio due to reference differences in the acquisition phase that could influence the registered amplitude in certain locations (due to different location of comparison references, for example: ears).
Third, because differences between noise acquisition in standard frequencies have shown to be negligible compared to noise introduced in higher bands (higher than 100 Hz) (Scheer et al., 2005), bandpass filters were selected both between 1 and 32 Hz, a good method to reduce external noise in the signal.
Four, Sample rate, bandwidth and resolution were similar in both acquisitions.
Five, another source of confounding is derived from the use of different software analysis, to avoid this issue, both samples were analyzed keeping statistical calculation parameters constant and avoiding software confounding effects (White, 2003).
Six, regarding the FFT power calculations, indeed, relative power is calculated based on the absolute power, as a measure of the percentage of each band over the total band range (1-32 Hz) and its use allows for natural correction of amplifiers and even differences in skull thickness (Thatcher et al., 2003). This calculation allows us to cancel the differences in gain or noise raised by differences in acquisition parameters.
For all these reasons, we think it is safe to say that acquisition parameters are thus compensated in case of differences in filter and gain parameters, external noise reduced to its minimum, and signal to noise ratio maximized. Overall, these control measures reduce the risk of external noise introduction when comparing between EEG samples. Maybe that is why, in recent years, the use of online available comparison databases/repositories in neuroscience measures is increasing. This is also the case for EEG samples (Johnstone & Gunkelman, 2003; Obeid & Picone, 2016; Thatcher et al., 2003). There is a long list of available repositories for clinical and research comparison either in MRI (The Brainomics/Localizer database), EEG (The European Epilepsy Database, EPILEPSIAE) or MEG (OpenNeuro). In our case, previous studies have utilized normative data bases to study differences between Fibromyalgia and non-clinical subjects for comparison of spectral absolute power, relative power and coherence. (https://doi.org/10.1177%2F155005941004100305) with good results.
We conclude that the probability to attribute differences between FMS and control sample to acquisition issues were minimized. But, even accepting some differences in amplitude due to calibration, the most important reason to accept the validity of the data is that calibration differences cannot explain the large differences between FM and control group, too large to be attributed to small differences in miss-calibration of the acquisition parameters, even once relative power band activity were calculated (with the exception to delta band).
In order to clarify this in the manuscript, we have added the following paragraph (lines 101-108), in the methods section,
“For the age-gender-matched control group, 23 healthy controls were randomly collected from a public standard database available on the Internet under the auspices of the Na-tional Center for Research Resources of the National Institutes of Health (16). This proce-dure of comparing Fibromyalgia EEG data with non-clinical population data from public repositories has been previously published with similar results (22). Gender distribution was of five males (21.8%) and eighteen females (78.2%) for both groups. T-test analysis in-dicated there were no significant differences in age (t (44)=.061, p=0.54).”
We included a paragraph with a brief comment in the limitation sections discussing the aforementioned issues (line 354-371).
“Because the use of the reference database is a relatively new, but increasingly adopted procedure (40,41), some possible limitations. It is debatable that the FM sample could not be completely comparable with the non-clinical subjects extracted from the standard da-tabase due to incompatibilities derived from different acquisition methods. However, sev-eral methodological techniques ensure the validity of this comparison, reducing acquisi-tion noise and sources of confusion (42). These methods include the use of averaging mounts, band-pass frequency filters that exclude frequencies above 100 Hz (43), and rela-tive power calculations to reduce differences individual measurements in skull thickness or amplifier calibration (44). These techniques show similar results between acquisition devices when calculating the FFT bands (45), as well as similar signal-to-noise ratios be-tween devices (up to 12 different ones), and even the fact that the comparison between modern low-cost EEG devices shows results comparable with those of medical grade for the calculation of frequency (46). These techniques and new research opportunities are opening up new avenues for the use of open repositories in neuroscience research and collaboration. Finally, this comparison procedure was previously validated by comparing FM patients with non-clinical subjects. Hargrove et al., (2010) found similar results of re-duced EEG power, high frequencies, and low coherence in FM patients compared to con-trol subjects (22).”
- The authors state that characterization of objective peripheral or central neurological structural deficits are needed to define fibromyalgia. However, fibromyalgia is a clinical syndrome that likely reflects a change in the function as compared to the structure of the nervous system. This is reflected by fibromyalgia being defined by rheumatology as compared to neurology.
RESPONSE:
We fully agree with the reviewer: Patients with FM present functional impairments (widely described in previous papers) but structural damages are not required to diagnose FM. Accordingly, the new version of the manuscript includes (lines 34-37),
“Despite the identification of these clinical features, current literature doesn´t report specific peripheral or central functional impairments that justify such symptoms, which difficulties the consensus on the diagnostic criteria, and hinders the identification of adequate treatment (4)”.
Furthermore, the hypothesis now reads as follows (line 87-89),
“The hypothesis is that patients with fibromyalgia present significant differences in EEG activity concerning controls and that this differential activity would be linked to a decrease in brain connectivity, as it happens in other diseases (13–15).”
- Due to concern for multiple testing, please at least list the pairs of electrodes that were evaluated prior to showing the electrodes with significant findings.
RESPONSE:
Thanks for the suggestion. We have added a table in the results section Table 3, presenting main fronto-temporal areas under the curve from ROC curve calculation using coherence derivations (lines 250-255).
|
Derivations |
Delta |
Theta |
Alpha |
Beta |
|
Fp1-Fp2 |
.390 |
.467 |
.294 |
.238 |
|
Fp2-T4 |
.843** |
.907** |
.875** |
.846** |
|
Fp1-T3 |
.843** |
.849** |
.884** |
.746* |
|
T3-T4 |
.580 |
.596 |
.706 |
.593 |
|
Fz-T3 |
.712* |
.684 |
.780* |
.765* |
|
Fz-T4 |
.799* |
.835* |
.774* |
.846** |
|
Fz-T3 + Fz-T4 |
.877** |
.794* |
.788* |
.822* |
|
Fp1-T3 + Fp2-T4 |
.765* |
.887** |
.913** |
.800* |
Table 3. Accuracy index - Area Under the Curve (AUC) for ROC curves calculated from Coherence values.*p<.05; **p<.001.
- The authors state in the abstract that they performed ROC analysis. The values from the ROC evaluate the predictions from a machine learning prediction algorithm and the ROC is not a method in itself. The authors should specify their machine learning methods in the abstract. In the main text, it seems like they only report a single amalgam feature as compared to developing a machine-learning model.
RESPONSE:
Thanks for the comment. Even though many papers report the use of machine-learning techniques to generate predictive algorithms (e.g. patients vs controls), this manuscript doesn´t describe a machine-learning model. Actually, results depict the differential distribution of the data displayed on a ROC curve. ROC curves are graphical representations of a binary classifier system as its discrimination threshold varied. From that point of view, it is a classical classificatory option. In order to state this difference clearly in the text, the new version of the manuscript incorporates more details for clarity purposes (lines 160-183),
“To elucidate whether the connectivity pattern may serve to identify patients with fibromyalgia, we calculated ROC curves through the SPSS ROC curve calculation implementation, obtaining sensitivity and specificity values and a total discrimination index. This index was used together with the clinical diagnosis to calculate the Sensitivity / Susceptibility S = TP / (TP + FN) and the Specificity E = TN / (FP + TN) of the EEG as a diagnostic tool, were TP regards “true positive”, TN means “true negative”, FN means “false negative”, and FP regards “False positive”. ROC curves were used as an accuracy index to explore the discriminative validity of EEG parameters Note that no machine learning methods were used. The discriminative capacity, following the ROC curve analysis, using 95% confidence intervals (CI) was calculated using the following formula: Sensitivity- (1-Specificity) (26). The software SPSS v.23 was used for statistical analysis.”
- Due to the likelihood for evaluating many potential amalgam features and only reporting one, with no discussion of cross-validation or training/testing groups, I have high concern for overfitting to this specific small dataset, and that these results will not generalize to a broader population of patients with fibromyalgia.
RESPONSE:
We would like to thank the reviewer for making us aware of that concern. With no doubt, our results would require a wider reproduction before being susceptible of clinical application. However, it is undoubtable that these results provide a fresh idea respect to the functional misbalances that may be at the basis of the clinical features in FM. More interestingly, they are in line with previous findings in FM resting state connectivity, as fronto-temporal connectivity alterations (Cifre et al., 2012; Fallon et al., 2016; Flodin et al., 2015; Harris et al., 2014), often associated with a decreased white-matter volume in FM subjects compared to controls in frontal regions (Cagnie et al., 2014).
- Cifre et al., 2012, https://journals.lww.com/psychosomaticmedicine/Abstract/2012/01000/Disrupted_Functional_Connectivity_of_the_Pain.10.aspx
- Fallon et al., 2016
https://doi.org/10.1371/journal.pone.0159198
- Flodin et al., 2015
https://doi.org/10.1016/j.nicl.2015.08.004
- Harris et al., 2014
https://pubmed.ncbi.nlm.nih.gov/24815079/
- Cagnie et al., 2014
https://doi.org/10.1016/j.semarthrit.2014.01.001
We have added a brief note about this concern in the limitation sections (lines 362-367),
“Alterations in functional connectivity are frequent in patients with FM, in particular alterations in fronto-temporal connectivity (Cifre et al., 2012; floding, 2015; Harris et al., 2014), often associated with a decrease of white matter volume compared to controls in frontal regions (Cagnie et al., 2014). Our results replicate these previous findings. However, since we have specifically studied fronto-temporal functional connectivity as a region of associations of interest (due to sample size limitations), these results should be viewed with caution.”
- The authors should provide a citation for the direct and indirect expenditures associated with FM when they discuss it in the introduction.
RESPONSE:
The new version of the manuscript includes specific details on the expenditures associated with FM, as seen in line 61-63,
“The mean annual cost per patient ranged US $2,274 to $9,573 or even more in various studies depended on the severity of symptoms and rout of cost calculation (5).”
7.1 The author’s initial hypothesis of “lower” EEG activity is quite vague and nonspecific. There is not a meaningful biological hypothesis for why the patient may have “lower” EEG activity.
RESPONSE:
Previous studies stated a decreased connectivity function on several neurological and psychiatric pathologies. Following that rationale, we hypothesized that pain symptoms in FM could related with the pain signal interpretation and/or with the signal integration. Thanks to the reviewer comment, we now noted that the expression “lower” is not precise enough, so the new version of the manuscripts presents differently the idea we wanted to transmit, as follows (line 72-89),
“Based on these data, some observations confirm that functional connectivity is generally impaired in FM patients (7-9). In particular, recent FM electroencephalography (EEG) studies during the resting state showed abnormalities in Theta (10) oscillation and a decrease in the power spectrum in the 9-14 Hz band in the central regions compared to controls (7). In agreement, another study found reduced lower frequencies and increased higher frequency activities in FM patients with respect to controls (8). Even other neuroimaging techniques, such as functional magnetic resonance imaging (fMRI), showed disrupted connectivity at the Theta frequency for FM patients compared to controls in de-fault network (9) and resting-state sequences (10). However, there is so little information about the electrical coherence between brain regions in FM patients compared to controls (11) and specific connections between regions aren´t addressed yet. The analysis of such differences might be based on a diagnostic method for FM using the EEG technique (12). In this research, we attempt to identify the electrical connectivity differences between patients with FM and matched control groups using EEG analysis and, specifically, studying the differences in functional connectivity patterns. The hypothesis is that patients with fibromyalgia present significant differences in EEG activity concerning controls and that this differential activity would be linked to a decrease in brain connectivity, as it happens in other diseases (13–15).”
7.2 The definition of global lower activity also is unclear, as attenuation of EEG voltage can be seen in many severe conditions including global hypoxic-ischemic damage and bilateral subdural or subgaleal injuries. This is differentiated from other changes in EEG caused by global cerebral dysfunction like encephalopathy, where the predominant frequency of EEG shifts slower, so there’s global slowing of the EEG.
RESPONSE:
The current version of the manuscript include further reference that sustain the hypothesis, providing enough justification to run the analysis on EEG data. In order to know more and get closer to understand in what way the EEG activity in FM patients differ from controls, we analysed the connectivity among the samples. This is an initial approach to know the differences. In our opinion, the key point here remains the meaning of such connectivity pattern differences. It is not just that FM patients complaint about pain, but that they show a differential functional pattern. Undoubtly, pathological hallmarks may be common to further clinical conditions. Further research should aim to describe the differences and the similarities among different syndromes.
- In the procedure section, the authors state “patients usually show the presence of areas with atrophy or signs of aging or irrigation alterations.” Based on my clinical experience and knowledge of the neuroimaging research in fibromyalgia, typical imaging findings for patients with fibromyalgia are normal. Therefore, the authors must provide a citation to support their claim.
RESPONSE:
Some references are now included, in order to sustain the connection between FM and brain atrophy (please, see line 130-135).
“The grapho-elements of pathological significance were registered (i.e. frequent posterior sharp-waves) (17) since the morphological mapping tests of these patients usually show the presence of areas with atrophy (18,19), with signs of aging (20) or irrigation alterations (21) (Figure 1). No pathological signals were found on the EEG registry of the control sample.”
- In the procedure, please provide a citation of the coherence method for analyzing functional connectivity.
RESPONSE:
The new version of the manuscript includes the requested citations, as follows (line 140-146),
“Quantitative analyses (both FFT and Coherence analysis) were per-formed using the Brainstorm Software (23) on MATLAB compiler runtime R2015b (MCR v9.0). The amplitude for each frequency band was calculated through the Fast Fourier Transformation (FFT) and the functional connectivity through the coherence method (https://neuroimage.usc.edu/brainstorm/) previously used (24). The localization of the ab-normal activity source required the Standardized low-resolution brain electromagnetic tomography (sLORETA) method (25).
- From Table 1, the authors report power in each spectral band, but they do not specify the location of the channels where power was calculated. Was this a weighted average across all channels with an average reference, or another combination of channels?
RESPONSE:
Thank you for noticing this detail. The measures were averaged across all channels with an average reference. This clarification is now included in the methods section, lines 139-140.
“Global EEG power was calculated with a weighted average across all channels with an average reference”.
- The authors state that the EEGs were visually evaluated and showed 17.8% spike type grapho-elements in patients with fibromyalgia. Please specify the expertise of the reader, as EEG reading is difficult and requires many years of clinical training.
RESPONSE:
The manuscript now clarifies the experience of the medical doctor who visually evaluated the registries (lines 128-130), as follows,
“A 50-years-experience in EEG medical doctor run a qualitative analysis by visual inspection with specific assemblies of bipolar montages”.
- Based on the term used, it’s difficult to know if these spikes are artefact or suggestive of other pathology, because the term used does not conform to the standards of the Clinical Neurophysiology Society.
RESPONSE:
As suggested by the reviewer, the sentence has been modified, as follows
(Lines 118-120),
“The grapho-elements of pathological significance were registered (i.e. frequent posterior sharp-waves) (17) since the morphological mapping tests of these patients usually show the presence of areas with atrophy (18,19), with signs of aging (20) or irrigation alterations (21) (Figure 1). No pathological signals were found on the EEG registry of the control sample.”
The reference is Tatum WO, Olga S, Ochoa JG, Munger Clary H, Cheek J, Drislane F, et al. American Clinical Neurophysiology Society Guideline 7. J Clin Neurophysiol [Internet]. 2016 Aug;33(4):328–32. Available from: http://journals.lww.com/00004691-201608000-00009
- The authors should show a tracing of the anomalous activity on EEG, because it’s only present in 18% of patients and it is unclear how to interpret it.
RESPONSE:
As requested, the current version of the manuscript incorporates a new figure (Figure 1) that shows an example of anomalous activity.
Figure 1. Example of EEG registry that presents abnormal activity (squared in grey).
- The number of matched controls is inconsistent between the methods and discussion. Please clarify this to ensure accuracy. There’s a marked difference between utilization of these methods with unbalanced groups (14 controls) and balanced groups (23 controls).
RESPONSE:
Typo. We totally agree with the reviewer and feel sorry for the error. The new version of the manuscript incorporates the corrections in order to clarify that both patients and control sample counted with 23 subjects each.
Minor comments:
- Spectral power is a logarithmic quantity, so the author should perform student’s t-tests on the log of spectral power in each frequency band, as compared to the absolute value.
RESPONSE:
Requested values are available on table 1. For a better understanding, the current version of the manuscript now includes the information in Table title (line 192-193), as follows,
“Table 1. Comparisons (t-test) of Absolute and Relative Mean Frequencies for FM and Control group *p<.001.
- When the authors reference fibrofog, they should include a citation on the topic at the end of the sentence.
RESPONSE:
The new version of the manuscript includes the specific information required (lines 34-54), as follows,
“On top of the classical pain that characterized FM patients, some cognitive impairments as loss of memory and attention (i.e. fibrofog, subjective experience of cognitive dysfunc-tion) start being recognized as independent symptom within the FM frame (2). Cognitive issues increase difficulties in adaptation, suffering, and disability, and have a high impact on the functionality of these patients (1,3).“
The included reference is,
Walitt B, Čeko M, Khatiwada M, Gracely JL, Rayhan R, VanMeter JW, et al. Characterizing “fibrofog”: Subjective appraisal, objective performance, and task-related brain activity during a working memory task. NeuroImage Clin [Internet]. 2016;11:173–80. Available from: http://www.ncbi.nlm.nih.gov/pubmed/26955513
- The tone of the introduction in the abstract and main text are more extreme than typically seen in formal writing about clinical conditions.
RESPONSE:
The new version of the manuscript includes some modifications in order to reduse the “extreme tone” referred by the Reviewer. The introduction in the abstract (lines 11-18) now reads as follows,
“The diagnosis of fibromyalgia (FM) doesn´t gather consensus even today, almost 40 years after the recognition of this disease. Recent findings have shown abnormal patterns in specific cortical areas of the central nervous system of patients with fibromyalgia. Based on EEG data, this study aims to describe the connectivity patterns in a group of patients to establish differential param-eters and compare them with a sample of healthy controls. Although previous studies demon-strated impaired patterns of brain activity during pain processing in patients with FM, altera-tions in spontaneous brain oscillations, in terms of functional connectivity or microstates, re-quire to be explore”.
- In the introduction, the authors should define the “Alpha-2” band as alpha has a standard definition but “Alpha-2” does not.
RESPONSE:
The new version of the manuscript includes the specific information required (line 72-76), as follows,
“In particular, recent FM electroencephalography (EEG) studies during the resting state showed abnormalities in Theta (10) oscillation and a decrease in the power spectrum in the 9-14 Hz band in the central regions compared to controls (7).”

Reviewer 4 Report
The present article elucidated EEG characteristics of patients suffering fibromyalgia by investigates differences in connectivity between healthy controls and patients. This is a very relevant topic that could have interesting application in the clinical practice. However, the methods section needs some clarifications, especially regarding the data collection and statistical methods. The language of the article needs to be substantially revised in several parts, especially in the introduction and the methods section. Abstract.
The author writes “ We registered the resting state….” However, it is my understanding that data was extracted from databases, so this should be clarified in the abstract.
Introduction
Line 39– the words “which difficulties” needs revising an the sentence is grammatically incorrect
line 45 – “depended on” should read “depending on”
lines 50 to 52 are difficult to understand, especially the words “by altering” .
line 57 – the paragraph starts stating “ Based on these data” , it is not clear what is the data the author refer to.
Line 64 – The author refer to a study reporting Theta frequency using fMRI. Frequency bands are extracted from EEG data so it is unclear how this information was extracted using fMRI.
Lines 57 to 74 – in this paragraph the authors discuss functional connectivity connectivity, but they only mention alterations in frequency bands and studies based fMRI. It would be useful to include literature on connectivity analysis based on EEG data as this is the method used in the article.
Line 74- the authors describe their hypothesis, says that same “…happens in other disease”. I would encourage the author to expand this aspect, mentioning the other disease and how brain activity is affected in that case, thus making the link the FM clear.
The introduction points out at the lack evidences on peripheral or central functional impairment that justify cognitive impairment in FM. However, the article does not contribute to this issue, as measurements of cognitive impairment were not included. In the discussion, the authors mention similarities with elderly population. Please add references to literature and expand the discussion of such similarities.
Methods
Line 92 – The authors mention that the data was previously published with similar results. Please explain what is meant by similar results.
Line 92 – Please explain what is meant by “normal characteristics” for the controls.
Line 96 - Was the t-test necessary since the sample was age matched? Otherwise, please, explain how the match was performed.
Line 96 – The sentence “ The Ethical Committee….” Is a repetition of line 82-83
Line 97 – 100- These lines are grammatically incorrect. In line 100 the word “ provided” should be deleted.
Line 102 – The sentence “served to the registry the data” seams grammatically incorrect.
Line 105 – the word “disposition” should be substituted with the word “placement”.
Line 106 – the sentence “a 50- years-experience in EEG medical doctor” seams grammatically incorrect, also it is unclear to me what is meant by “visual inspection with specific assemblies of bipolar montage”
Please provide more information about how the data was collected. What about ground and reference for the recordings? Were the patients sitting? Did the recording include only 15 mins eyes-closes? Where all 15 minutes included in the data analysis?
Figure 1 is of poor quality. It is not possible to read the labels nor the units used.
Line 126 – the authors refer to qualitative variables. Which are these variables?
Line 128 – Were multiple T-test executed? Would this not increase the chance of error? Where any correction applied in in this case?
Line 142 – “were” should read “where”
Line 144 . A “.” Is missing
Results.
Line 150- 154 – should be in the Methods section.
Table 1 refers to relative frequencies. How were these calculated?
Line 168 – Consider deleting the words “statistical figures”.
Figure 2 – Units are missing from the y axes.
Line 172-173 – “…frequency maps of the FM sample showed greater activity in the right parietal region… with respect to the rest of the bands”, did the author mean with respect of the rest of the regions?
Figure 3 – FMS should read FM.
Discussion
Line 282 – “fibbers” should read “fibers”
Author Response
Please, see the attached doccument.

Reviewer 5 Report
Discussion of how similar EEG measurements appear in diseases with overlapping symptoms should be discussed. This will be important for the specificity of potential EEG tests for FM.
Presentation of the discriminatory power of these measurements can be improved by addition of specificity and sensitivity values next to the corresponding AUC in Table 2's ROC analysis.
Author Response
Please, see the attached doccument.
Round 2
Reviewer 3 Report
I appreciate the efforts of the authors and their interest in the topic, which I think is quite worthy of examination. The authors have conducted an appropriate investigation, but on the wrong data. The perfect confound of acquisition location with diagnosis is not a limitation that can be addressed based on the authors' response and is not something that is overcome by clearly discussing this as a limitation or avenue for future investigation in the discussion.
Major comments:
- The confound of acquisition location. The authors correctly state that there are some differences in technical acquisition and referencing protocols that do not influence the signal or processing of EEG information. The authors do not address other parameters including but not limited to unmeasured differences in electrode location placement despite similar guidelines (there is marked variation in this when measured); differences in the ambient environment of recording including conditions of eyes open, eyes closed, distractions from other light, noise, and olfactory sources; and differences in electrical shielding protocols between neurophysiology labs. These factors can have large impacts on networks like the default mode network, therefore it’s feasible that these and other unmeasured and unmeasurable confounds can have a large impact on connectivity.
- I appreciate and participate in the development of the large multisite research and clinical databases for neurodiagnostic information. However, a key difference between these databases and the current study is that site is not a perfect confound with diagnosis. Said another way, each site provides both patients with and without FM. In that way, the control subjects can be used to assess statistically for variation across site. The ENIGMA neuroimaging group has some great recent manuscripts on the various protocols for data harmonization across site and the numerous sources of variation that can be site specific, as well as how their analysis protocols address these variations.
Minor comment:
- As a point of sensitive language, I believe it’s appropriate to use the phrase “patients with FM” as compared to “FM patients.” This way we define participants as people/patients first that then have the disorder.
Author Response
Madrid (Span), May 26th, 2021
Reviewer 2
Special Issue Editor, Journal of Clinical Medicine
Dear Reviewer,
We enclose the updated version (version 3) of the manuscript entitled “Fibromyalgia diagnosis based on EEG connectivity patterns” for consideration as an article in the Special Issue "Chronic Fatigue Syndrome/Myalgic Encephalomyelitis: Diagnosis and Treatment" (Clinical Neurology Section, Journal of Clinical Medicine).
First, we are deeply thankful for the received consideration and time invested in reviewing our manuscript. Please, find below our responses to the second peer review.
Query 1. The confound of acquisition location. The authors correctly state that there are some differences in technical acquisition and referencing protocols that do not influence the signal or processing of EEG information. The authors do not address other parameters including but not limited to unmeasured differences in electrode location placement despite similar guidelines (there is marked variation in this when measured); differences in the ambient environment of recording including conditions of eyes open, eyes closed, distractions from other light, noise, and olfactory sources; and differences in electrical shielding protocols between neurophysiology labs. These factors can have large impacts on networks like the default mode network, therefore it’s feasible that these and other unmeasured and unmeasurable confounds can have a large impact on connectivity.
RESPONSE: We thank the reviewer for sharing his/her concerns about the placement of electrodes, light, noise, and smell sources. Indeed these measurements can influence the EEG register and cause artifacts. Just to clarify, only eyes closed registries were included in this study. The differences in electrode placement were minimized by the use of a global average index. Regarding environmental differences during the EEG acquisition, it is common protocol to register in complete silence, in a comfortable and isolated room free from unpleasant stimuli. We did follow these common practice suggestions, and now explicitly state these factors and their importance in the description of the acquisition protocol in the procedure/methods section.
These potential confounders are inevitable to a certain extent, some confounds are measurable and others are unmeasurable, as the reviewer states, but this is also true for every experiment in Science and does not necessarily imply any major risk for the validity of our experiment. On the one hand, noise, light distractors, etc. generate specific artifact patterns (ie. movement, eyes artifacts) identifiable for an expert electroencephalographic of 50 years of experience.
On the other hand, these potential confounders not only depend on light, sound, and smell but the person itself. In other words, compared to having FM, light, sound, and smell should be regarded as negligible. In brain imaging methods, the fact that a pattern results from the analysis is due to the persistent presence of a specific signal flow. The connectivity patterns extracted from the EEG are the result of a persistent brain signal. We are more inclined to conclude that differences between samples are not due to these secondary factors but are specific FM connectivity patterns, being the control sample a valid one. Indeed, this is the opinion of other researchers that have used the same database for comparison with sleep disorders or autism, among others (see the response to query 2). Furthermore, as stated in the discussion section, these altered connectivity patterns are in line with previous studies. We think that these coincidences with other studies support the validity of the comparison sample. This explanation is now included in the discussion section.
Included in the text (lines 249-259):
“ Going deeper in the analysis of the affected bands, the most severe decrease took place in Alpha, Theta, and Beta frequency bands and was not significant in the Delta band. On one hand, given that the first three bands depend on the cortical neuronal inter-action and its networks, it is reasonable to link these alterations with the morphological impairments earlier described. On the other hand, Delta activity is more dependent on the somas of the deep pyramidal neurons affected in degeneration, usually by irrigation defects (40). Other studies have reported similar differences between patients with FM and healthy controls while solving problems of various types. In Global Field Power analysis, patients with FM presented lower modulation of Alpha and Theta, less synchronization, and lower spectral density, which indicates the presence of excessive neuronal noise (41) “
References
Choe, M. K., Lim, M., Kim, J. S., Lee, D. S., & Chung, C. K. (2018). Disrupted Resting-State Network of Fibromyalgia in Theta frequency. Scientific reports, 8(1), 2064.
Fallon, N., Chiu, Y., Nurmikko, T., & Stancak, A. (2018). Altered theta oscillations in resting EEG of fibromyalgia syndrome patients. European journal of pain, 22(1), 49-57.
Fallon, Nicholas, Chiu, Y., Nurmikko, T., & Stancak, A. (2016). Functional connectivity with the default mode network is altered in fibromyalgia patients. PloS one, 11(7), e0159198.
González-Roldán, A. M., Cifre, I., Sitges, C., & Montoya, P. (2016). Altered dynamic of EEG oscillations in fibromyalgia patients at rest. Pain Medicine, 17(6), 1058-1068.
One last thought. Thinking these subject through carefully, to say that environmental factors could compromise the comparison between EEG samples or that databases can not be used properly, indirectly would imply that the highest level of abstraction methodological techniques oriented to approximate to a "universal representative mean", which are meta-analysis, would be impossible to perform in EEG because it would be impossible to compare. This is not the case, as, in a quick review of the Pubmed literature database, we found that, only in the last 5 years (from 2016 to 2021), two hundred and twenty-two meta-analyses (221 meta-comparison studies) have been published. Fortunately, it seems that the comparison between EEG samples is feasible, and can help the scientific community to provide useful insights taken from all the studies altogether. Overall, the research hypothesis and the results presented in our study are interesting enough to be later replicated or falsifiable for other researchers.
Query 2. I appreciate and participate in the development of the large multisite research and clinical databases for neurodiagnostic information. However, a key difference between these databases and the current study is that site is not a perfect confound with diagnosis. Said another way, each site provides both patients with and without FM. In that way, the control subjects can be used to assess statistically for variation across sites. The ENIGMA neuroimaging group has some great recent manuscripts on the various protocols for data harmonization across the site and the numerous sources of variation that can be site-specific, as well as how their analysis protocols address these variations.
RESPONSE: We thank the reviewer for pointing at such relevant aspects of multi-site studies. As he/she states, the ENIGMA group has addressed them in some recent papers. Even though we are conscious that this type of comparison has its limitations, previous studies have applied both the same design and the same database as us: EEG recordings from control subjects (PhysioBank database) compared to EEG local recordings from patients with epilepsy (Ravan & Begnaut, 2019; Piangereli et al., 2018) and patients with autism spectrum disorders (Jadhav et al., 2014). There are even publications comparing two different databases -control vs sleep disorders patients- (Sharma et al., 2021; Dimitriadis et al., 2018) or two sleep disorder patients databases (Fu et al., 2021). Additionally, some authors have verified the suitability of site/device/experimental paradigms comparisons in published research studies, the so-called “EEG mega-analysis” (Bigdely-Shamlo et al., 2020).
It is also remarkable that the relative frequencies described in the control group perfectly fit in the standard values (Niedermeyer, E. (1999), allowing us to assume that the comparisons between this control sample and the sample of patients with fibromyalgia are perfectly reasonable.
EEG has the advantage of being harmless to the patient, susceptible to repeat, and facilitates important functional information at the cost of being difficult to interpret and time-consuming technique. Fortunately, these inconveniences may be reduced with the use of comparative online EEG repository resources, which are increasing in recent years. They are theoretically doable and might accelerate the research performed using EEG.
References:
Bigdely-Shamlo, N., Touryan, J., Ojeda, A., Kothe, C., Mullen, T., & Robbins, K. (2020). Automated EEG mega-analysis I: Spectral and amplitude characteristics across studies. 15(207), 116361.
Dimitriadis, S. I., Salis, C., & Linden, D. (2018). A novel, fast, and efficient single-sensor automatic sleep-stage classification based on complementary cross-frequency coupling estimates. Clinical Neurophysiology, 129(4), 815–828. https://doi.org/10.1016/j.clinph.2017.12.039
Jadhav, P. N., Shanamugan, D., Chourasia, A., Ghole, A. R., Acharyya, A., & Naik, G. (2014). Automated detection and correction of eye blink and muscular artifacts in EEG signal for analysis of Autism Spectrum Disorder. 2014 36th Annual International Conference of the IEEE Engineering in Medicine and Biology Society, 1881–1884. https://doi.org/10.1109/EMBC.2014.6943977
Piangerelli, M., Rucco, M., Tesei, L., & Merelli, E. (2018). Topological classifier for detecting the emergence of epileptic seizures. BMC Research Notes, 11(1), 392. https://doi.org/10.1186/s13104-018-3482-7
Ravan, M., & Begnaud, J. (2019). Investigating the Effect of Short Term Responsive VNS Therapy on Sleep Quality Using Automatic Sleep Staging. IEEE Transactions on Biomedical Engineering, 66(12), 3301–3309. https://doi.org/10.1109/TBME.2019.2903987
Sharma, M., Tiwari, J., & Acharya, U. R. (2021). Automatic Sleep-Stage Scoring in Healthy and Sleep Disorder Patients Using Optimal Wavelet Filter Bank Technique with EEG Signals. International Journal of Environmental Research and Public Health, 18(6), 3087. https://doi.org/10.3390/ijerph18063087
Fu M, Wang Y, Chen Z, Li J, Xu F, Liu X, et al. Deep Learning in Automatic Sleep Staging With a Single Channel Electroencephalography. Front Physiol [Internet]. 2021 Mar 3;12. Available from: https://www.frontiersin.org/articles/10.3389/fphys.2021.628502/full
Niedermeyer, E. (1999). The normal EEG of the waking adult. Electroencephalography: basic principles, clinical applications and related fields, 20(4), 149-173.
Query 3. Minor comment: As a point of sensitive language, I believe it’s appropriate to use the phrase “patients with FM” as compared to “FM patients.” This way we define participants as people/patients first that then have the disorder.
RESPONSE: We agree and the manuscript now incorporates those changes.
Yours sincerely,
Cristina Nombela Otero, PhD
Biological and Health Psychology, School of Psychology
Autonomous University of Madrid (UAM), Madrid (Spain), PC: 28043
Phone number. +34 91 4975921
Cristina.nombela@uam.es